# Autologous hGMSC-Derived iPS: A New Proposal for Tissue Regeneration

**DOI:** 10.3390/ijms25179169

**Published:** 2024-08-23

**Authors:** Ylenia Della Rocca, Francesca Diomede, Fanì Konstantinidou, Valentina Gatta, Liborio Stuppia, Umberto Benedetto, Marco Zimarino, Paola Lanuti, Oriana Trubiani, Jacopo Pizzicannella

**Affiliations:** 1Department of Innovative Technologies in Medicine & Dentistry, “G. d’Annunzio” University of Chieti-Pescara, Via dei Vestini, 31, 66100 Chieti, Italy; ylenia.dellarocca@unich.it; 2Department of Psychological Health and Territorial Sciences, “G. d’Annunzio” University of Chieti-Pescara, Via dei Vestini, 31, 66100 Chieti, Italy; fani.konstantinidou@unich.it (F.K.); valentina.gatta@unich.it (V.G.); liborio.stuppia@unich.it (L.S.); 3Center for Advanced Studies and Technology (CAST), “G. d’Annunzio” University of Chieti-Pescara, Via Luigi Polacchi 11, 66100 Chieti, Italy; paola.lanuti@unich.it; 4Department of Cardiac Surgery, “S.S. Annunziata” Hospital, ASL 2 Abruzzo, Via dei Vestini, 66100 Chieti, Italy; umberto.benedetto@unich.it; 5Department of Neuroscience, Imaging and Clinical Sciences, “G. d’Annunzio” University of Chieti-Pescara, Via Luigi Polacchi 11, 66100 Chieti, Italy; 6Department of Cardiology, “S.S. Annunziata” Hospital, ASL 2 Abruzzo, Via dei Vestini, 66100 Chieti, Italy; m.zimarino@unich.it; 7Department of Medicine and Aging Science, “G. d’Annunzio” University of Chieti-Pescara, Via dei Vestini, 66100 Chieti, Italy; 8Department of Engineering and Geology, “G. d’Annunzio” University of Chieti-Pescara, Viale Pindaro, 42, 65127 Pescara, Italy; jacopo.pizzicannella@unich.it

**Keywords:** hGMSCs, iPS cells, EBs, EVs, regenerative medicine

## Abstract

The high mortality in the global population due to chronic diseases highlights the urgency to identify effective alternative therapies. Regenerative medicine provides promising new approaches for this purpose, particularly in the use of induced pluripotent stem cells (iPSCs). The aim of the work is to establish a new pluripotency cell line obtained for the first time by reprogramming human gingival mesenchymal stem cells (hGMSCs) by a non-integrating method. The hGMSC-derived iPS line characterization is performed through morphological analysis with optical and electron scanning microscopy and through the pluripotency markers expression evaluation in cytofluorimetry, immunofluorescence, and RT-PCR. To confirm the pluripotency of new hGMSC-derived iPS, the formation of embryoid bodies (EBs), as an alternative to the teratoma formation test, is studied in morphological analysis and through three germ layers’ markers’ expression in immunofluorescence and RT-PCR. At the end, a comparative study between parental hGMSCs and derived iPS cells is performed also for the extracellular vesicles (EVs) and their miRNA content. The new hGMSC-derived iPS line demonstrated to be pluripotent in all aspects, thus representing an innovative dynamic platform for personalized tissue regeneration.

## 1. Introduction

The growing number of chronic diseases afflicting the global population underscores the urgency of identifying alternative efficient technologies. In this context, regenerative medicine offers solutions. Regenerative medicine is a branch of medicine supported by multidisciplinary and translational research that is rapidly evolving thanks to the use of innovative technologies for the repair and replacement of damaged cells, tissues, and organs [1]. The core of regenerative medicine is stem cells [2], characterized by self-renewal and the ability to differentiate into specialized cell types. Embryonic stem cells (ESCs), given their pluripotency, would be the best solution for cell therapy in new regenerative treatments. Research on ESCs has continued to spark moral, ethical, and legal debates related to their use. For this reason, research in the regenerative field has focused on the study and use of multipotent adult stem cells (MSCs). MSCs are a heterogeneous population of cells with fibroblastoid morphology and specific characteristics [3]. The so-called minimum criteria of MSCs are still valid and used today and include (i) the ability to self-renew; (ii) multipotency with osteogenic, chondrogenic, and adipogenic potential; and (iii) the expression of characteristic surface markers, such as the cluster differentiation (CD)73, CD90, and CD105, while lacking the expression of CD14, CD34, CD45, and human leukocyte antigen-DR (HLA-DR) [4]. Moreover, MSCs show plastic-adherent growth and are expandable in vitro over several passages [5]. Depending on their tissue derivation, MSCs can have different developmental origins with specific transcriptomic characteristics and, thus, different biological properties and functions, such as differentiation potential, secretome composition, and immunomodulatory functions [6]. The gengiva is a good source for finding MSCs, since it is easier to access and is abundant, allowing us to obtain human gingival mesenchymal stem cells (hGMSCs) [7]. Furthermore, in recent years, to have a greater potential of the cells for therapeutic purposes and, at the same time, to bypass the controversies related to the ethical problems of ESCs use, a lot of studies are focusing on the generation of induced pluripotent stem cell lines (iPSCs) [8]. iPSCs are pluripotent stem cells that are generated directly from adult somatic cells. Induction to pluripotency is achieved through the forced expression of pluripotent stem cell-specific transcription factors by various reprogramming methods [9]. The pluripotency genes that characterize embryonic cells are Oct4, Sox2, and Nanog. Therefore, the cocktail used to perform reprogramming is a mix of these three transcription factors, which form a circuit for pluripotency that is self-regulator: they regulate each other’s expression. It has been seen that Oct4, Sox2, and Nanog can activate the genes needed to maintain pluripotency and to repress differentiation-specific transcription factors, maintaining the pluripotent state [10,11]. The iPSCs can be generated through different methods: nuclear transplantation, cell fusion, reprogramming by cell extracts, and direct reprogramming through gene manipulation [12]. Among the direct reprogramming methodologies, we can make a further distinction between integrating vector methods and genomic non-integrating methods [13]. These two types of direct reprogramming are the most widely used methods to generate iPSCs today [14]. Bibliographic studies show that not all iPSs obtained with different reprogramming methods are capable of leading to the formation of the embryoid body and to generate the three germ layers [15]. iPSC technology has created new candidates for disease modeling, drug screening, cell therapy, and regenerative medicine. The iPSCs will constitute patient-specific in vitro disease models, resulting in more reliable and translatable data in clinical approaches. iPSCs have brought forth important advances in research, but some limitations, including immunogenicity, remain in regard to their use in clinical practices [16]. For this reason, it is essential to study the vesicular products of stem cells’ secretome as potential therapeutics that bypass the problems related to the cell therapies [17]. Extracellular vesicles (EVs) are among the most studied secretome products, especially for regenerative purposes. EVs are small membranous particles of different sizes that are released from cells in different ways. The term “EVs” is collective and indicates different types of membranous particles of cellular origin that have been named according to biogenesis, morphology, and functional characteristics [18]. Once released, EVs generally act as paracrine factors: their content is selectively internalized by nearby or distant target cells, influencing their behavior. The EVs’ content is mainly lipids, proteins, and RNA. Surface proteins are critical for targeting ability; consequently, a modification of proteins on the surface can improve recognition of EVs. EVs have several advantages: small size, low immunogenicity, long-circulating half- life, good penetration, and good biocompatibility [19]. These characteristics make exosomes one of the best approaches for therapeutic alternatives. They represent a safe alternative to stem cell therapy involving the administration of whole cells [20].

The aim of this study was the characterization of a new pluripotency cell line obtained for the first time by reprogramming human gingival mesenchymal stem cells (hGMSC-derived iPS cell line) through one of the most innovative reprogramming methods without viral vector: StemRNATM 3rd Gen Reprogramming Kit. Since, albeit to a small extent, some pluripotency genes, such as Oct4, are already expressed in hGMSCs, a comparative study was set up between hGMSCs starting cells and hGMSC-derived iPS in terms of mesenchymal and pluripotency markers expression. Therefore, since hGMSC-derived iPS are generated from multipotent cells, and EVs derived by hGMSCs have regenerative properties, the EVs could carry different potentials. Therefore, we characterized the exosome content of both starting hGMSCs and hGMSC-derived iPS in order to identify different therapeutic approaches for regenerative medicine. Following the results obtained from the analysis of the exosomal microRNAs content, a comparative study between EVs obtained from hGMSCs and those derived from hGMSC-derived iPS was set up. 

## 2. Results

### 2.1. hGMSCs Characterization

hGMSCs were characterized by means of the International Society for Cellular Therapy criteria to define their multipotent mesenchymal profile. hGMSCs are able to differentiate into osteogenic and adipogenic lineages, and they were maintained under proper culture conditions for 21 and 28 days, respectively. The cells were positive for adipo Oil Red that evidence the lipid vacuoles (Appendix A). Cells were positive for Alizarin Red staining at the cytoplasmic level and calcium deposits (Appendix A). Undifferentiated cells were adherent on a plastic substrate with a typical fibroblast-like morphology (Appendix A). Differentiation ability was confirmed to evaluate the expression of the related osteogenic and adipogenic genes. In osteogenic committed cells, RT-PCR data showed an increase in the mRNA expression of RUNX-2 and ALP compared to the undifferentiated cells (Appendix A). FABP4 and PPARγ expression was increased in the adipogenic differentiated samples when compared to the basal cells (Appendix A).

### 2.2. hGMSCs and hGMSC-Derived iPS Characterization by Flow Cytometry

A large panel of markers (as reported in the method section) was used to confirm the phenotype of MSC and the pluripotency of iPSCs. hGMSCs clearly express CD29, CD44, CD73, CD90, CD105, and CD146, as also demonstrated by the related MFI values (Figure 1A). Furthermore, neither hGMSCs nor hGMSC-derived iPSs express leukocyte (CD45) or endothelial cell (CD31) markers, and they do not express HLA-DR, as established (Figure 1B). As shown in Figure 1C, hGMSC-derived iPS and hGMSCs were paralleled for the expression of typical pluripotency markers (NANOG, OCT3/4, SOX2, and SSEA4) that were overexpressed by hGMSC-derived iPS, as also evidenced by the related MFI ratio values. Furthermore, differences in CD expression in hGMSCs and hGMSC-derived iPS were analyzed. The data showed clear expression changes for CD29, CD44, CD73, CD90, CD105, CD146, CD166, and CD117. All of these last markers listed were higher in hGMSCs than hGMSC-derived iPS, with the exception of CD117, which was higher in hGMSC-derived iPS (Figure 1D).

### 2.3. hGMSC-Derived iPS Colonies Formation

On day 2 of the reprogramming process, hGMSCs’ pluripotency features were evident. The reprogrammed hGMSCs showed a round-shaped morphology when compared to the hGMSCs maintained in basal conditions that showed a fibroblast-like structure (Figure 2A,B). On the eighth day of reprogramming the first clusters began to form, becoming definitive colonies on the thirteenth/fourteenth day, where the colonies under the optical microscope had bright edges, in clear separation from the adjacent cells that were not reprogrammed and so remained multipotent (Figure 2C,D). Through an optical microscope, it is possible to appreciate the morphology of hGMSC-derived iPS, which is completely similar to that of embryonic cells, i.e., a very high nucleus/cytoplasm ratio given the poor cytoplasmic component compared to the nucleus, a typical feature of pluripotent cells (Figure 2E,F). This was more evident under the scanning electron microscope, highlighting the ultramorphological structure of the iPSCs obtained. The iPSCs, as a characteristic completely comparable to ES, in culture did not occupy the surface of the entire plate, but they formed clusters which, isolated by picking, allowed for the expansion of the colonies. From the starting well, 190 colonies in total were isolated. The morphology of the colonies formed by the hGMSC-derived iPS was evident under the optical microscope and even more under SEM, where the iPSCs with the classic pluripotent morphology of the high nuclei/cytoplasm ratio (Figure 2G,H) came into contact with each other within the same colony (Figure 2I–L).

### 2.4. Karyotype Reconstruction

The karyotype analysis showed chromosome stability for the presence of normal female karyotype in both hGMSCs and hGMSC-derived iPS cultures (Appendix A).

### 2.5. Pluripotency Markers Expression Analysis with Confocal Microscopy

The pluripotency markers Oct4, Nanog, and Sox2 were analyzed through immunofluorescence with confocal microscopy. In the hGMSC sample, there was no signal for Oct4, Nanog, and Sox2 (Figure 3). The pluripotency markers’ expression was not detected in hGMSCs, the starting cell line for reprogramming. The immunofluorescence for actin and nuclei highlights, first, the morphology of colonies in hGMSC-derived iPS samples, with the perimetral actin-like embryonic cells. Sox2 is highly expressed in hGMSC-derived iPS (Figure 3A), as well as Oct4 and Nanog (Figure 3B).

### 2.6. DNA Finger Printing

A multiplex assay analysis for short tandem repeats (STRs) was used to compare the allelic profiles of 17 Y-STR loci between the parental and the reprogrammed lines and resulted in authentication regarding the derivation of the reprogrammed line from the parental one (Appendix A–C).

### 2.7. qPCR

The data obtained from the real-time PCR analysis regarding the expression of pluripotency genes showed a significant increase in the expression of Nanog in hGMSC-derived iPS compared to the starting hGMSC cell line that showed a lower expression of the marker, which seemed even less expressed in the EB sample. Sox2 was highly expressed in iPS-derived hGMSCs, while it was poorly expressed in hGMSCs and not expressed in EBs. The expression of the POU5F1 gene was low in the hGMSC sample, increased significantly in hGMSC-derived iPS, and not detectable in EBs (Figure 4A). The analysis of the three embryonic layers’ genes’ expression demonstrated the ability of the iPS used for the formation of EBs to give rise to the three embryonic layers. A high expression of the ectodermal differentiation marker TUBB3 was demonstrated in the EBs sample, it was expressed less in hGMSCs, and it was not expressed in hGMSC-derived iPS. Endodermal differentiation was highlighted by a high expression of SOX17 in EBs, compared to hGMSCs and hGMSC-derived iPS, in which it was not assessable. The mesodermal marker T was highly expressed in EBs compared to hGMSCs and hGMSC-derived iPS (Figure 4B).

### 2.8. Analysis of EBs Formation with Optical Microscopy

After hGMSC-derived iPS were plated on a U-bottom plate; the cells’ aggregation was evidence (Figure 5A). On day 2, the bodies resulted in havinh a spheric structure with a clear and well-defined edge (Figure 5B). As the days went by, the EBs were not only enlarged in diameter but also in intensity, with a darker core compared to the surrounding lighter and less dense areas (Figure 5C). After 16 days, the EB is completely formed, with a morphology that is not always perfectly spherical. The edges are dark and well defined compared to the surrounding area (Figure 5D). In EB samples that were transferred to a flat-well plate to assess the migrating population from the EBs to the plate, cells began to migrate from the EB as early as 24 h after transfer (Figure 5E). In four days, the cells at the end of the migrating population create a distinct outline (Figure 5F red arrow). In the population of migrated cells, a distinction into three areas differing in morphology and cellular organization was evident (Figure 5G). The distinction of three different morphologies between the cells within the migrating population from EB remained evident until the end of sixteen days of culture (Figure 5H).

### 2.9. Histochemistry Analysis of EBs in Toluidine Blue

To evaluate the EBs morphology, the sections stained with toluidine blue were used. The obtained data showed the differences in morphology and in cellular organization, as already highlighted in the migrating population with optical microscopy. In these EB sections, it is possible to distinguish a potential trilineage: ectodermal tissue in neuroepithelial structures (Figure 5I, yellow arrow; Figure 5J, yellow circle); mesodermal tissue with connective morphology and connective fibers (Figure 5I, green arrow; Figure 5K, green arrow; Figure 5L, green arrow); and endodermal tissue in structures like the lining epithelia (Figure 5I, red arrow; Figure 5M, red circle).

### 2.10. Three Embryonic Layers’ Markers’ Expression Analysis with Confocal Microscopy

The data obtained from confocal analysis for the expression of the three embryonic layers’ markers in EB 3D samples showed the differentiation of EBs cells in ectodermal, mesodermal, and endodermal lines (Figure 6A1–A5,B1–B5,C1–C5). The ectodermal Tuj1 was highly expressed in 3D EB samples (Figure 6A1,A2). Immunofluorescence results for Tuj1 expression in EBs sections have been demonstrated the neuroepithelial nature of the structure. In EB section Tuj1 signal was high and localized in a round-shaped area (Figure 6A3,A4). The fluorescent signal of mesodermal marker αSMA was evident in specific areas of 3D EB during the z-stack analysis, which allowed for the reconstruction of the complete 3D EB structure (Figure 6B1,B2). The data obtained from αSMA immunofluorescence in 3D EB were confirmed by αSMA immunofluorescence in the EB’s section (Figure 6B3,B4). The 3D EB resulted in being positive for endodermal marker SOX17 in immunofluorescence in specific zones during the z-stack analysis, through which the reconstruction of the complete 3D EB structure was performed (Figure 6C1,C2). In the EB section, the endodermal differentiation was confirmed by SOX17 immunofluorescence detection (Figure 6C3,C4). The immunofluorescence analysis for Tuj1, αSMA, and SOX17 provided the markers’ localization inside the single cells and in the migrating population. The ectodermal Tuj1 was highly expressed in EB samples at a cytoplasmic level (Figure 6A5). Some Tuj1-positive cells resulted in being associated with each other, and the actin showed a morphology that resembled that of neuronal cells, confirming organization of neuroectodermal tissue within the 3D EBs and EB’s section. The mesodermal marker αSMA appeared strongly positive in migrating cell population from EBs. An αSMA signal was localized in the abundant cytoplasm and signal that followed the cytoskeleton morphology that may resemble that of myoblasts (Figure 6B5). The endodermal marker SOX17 was expressed in EB migrating cells with a nuclear localization. The actin showed a perimetral localization with a floor-like and polygonal morphology that could be similar to that of endothelial cells (Figure 6C5).

### 2.11. hGMSCs and hGMSC-Derived iPS EVs Flow Cytometry

The EVs obtained from hGMSCs and hGMSC-derived iPS were identified by flow cytometry, using a recently optimized method [21]. As shown in Figure 4, MSC-derived EVs express CD90, as required [21,22]. Both MSC- and iPSC-derived EVs were analyzed for the expression of typical EV markers (i.e., tetraspanins—CD63, CD81, and CD9), and their concentrations were obtained using the already reported volumetric count protocol [21]. The results demonstrated that iPSC-derived EV concentrations were higher (3743.9 + 568.2) than iPSC-derived EVs (53.8 + 15.0). Furthermore, the typical EV markers’ expressions were different: the CD63 and CD9 MFI ratios were higher in the EVs obtained from hGMSC-derived iPS than hGMSC EVs; CD90 was greatly decreased in hGMSC-derived iPS EVs compared with hGMSC EVs, while CD90 and CD81 were higher in hGMSC EVs than hGMSC-derived iPS EVs (Appendix A).

### 2.12. miRNA Characterization and Analysis

The real-time analysis evidenced the presence of 80 microRNAs, characterized in the EVs of both the hGMSC and hGMSC-derived iPS groups (Appendix A). On the other hand, 66 microRNAs were detected exclusively in the EVs of the parental line (Appendix A), and 8 only in ones of the reprogrammed: hsa-miR-122, hsa-miR-138, hsa- miR-142-3p, hsa-miR-143, hsa-miR-152, hsa-let-7b, mmu-miR-451, and hsa-miR-486 (Appendix A). The expression levels of 41 of the 80 shared microRNAs were found to be differentially modulated in the EVs of reprogrammed compared to the parental line (0.7 < fold-change < 1.2, *p*-value < 0.0001, <0.001, < 0.01 and < 0.05) (Appendix A, Figure 7A,B).

### 2.13. Ingenuity Pathway Functional Analysis

The ingenuity pathway analysis performed on the characterized microRNAs of both the parental- and the reprogrammed-derived EVs showed that they were involved in a series of important biological processes. As far as the microRNAs contained solely in the parental EVs were concerned, they were found to actively participate in gene expression, cellular movement, growth and proliferation, inflammation, and cell death and survival (Figure 7C). Regarding the eight microRNAs detected in the EVs of the reprogrammed group, these were involved in similar processes, with the addition of hematopoiesis and cell-to-cell signaling and interaction (Figure 7D).

### 2.14. miRNAs Expression Profile

Forty-one microRNAs resulted in being significantly modulated between the two groups. An additional Ingenuity Pathway Analysis was performed on these microRNAs, highlighting the main functions and biological processes they also participated (shown in Figure 7E). Following the analysis, processes such as inflammatory disease and response; endocrine and metabolic disorders; cellular development, movement, death, and survival, as well as cellular response to therapeutics were equally evidenced.

### 2.15. Ingenuity Pathway Network Analysis

An IPA was also used to highlight the networks in which the characterized and differentially modulated microRNAs are involved. For microRNAs solely present in the paternal EVs, two networks were generated with scores ranging from 45 to 18 for the top network, centered on the central node gene HNF4A (Figure 7F) and 39 to 16 for the second one, centered on gene TP53 (Figure 7G). 

For microRNAs exclusively present in the reprogrammed-derived EVs, on the other hand, a single top network was generated, also centered on gene TP53 (Figure 7G), with a score ranging from 21 to 7.

## 3. Discussion

The data from immunofluorescence and real-time PCR analysis for pluripotent genes showed positive results for the expression of OCT4, SOX2, and NANOG in hGMSC-derived iPS colonies, confirming the pluripotency of this new cell line. The results shown in this study demonstrate the pluripotency of a newly characterized iPS cell line, hGMSC-derived iPS, obtained by non-integrative reprogramming, starting from hGMSCs and not from classical fibroblasts. The DNA finger-printing analysis indicates the effective derivation of this new iPS cell line from the parental hGMSCs. The new hGMSC-derived iPS cell line proved to be stable in the karyotype without chromosomal aberrations. This makes it evident that reprogramming through a non-integrative method, with the use of reprogramming cocktails delivered by Lipofectamine, did not affect the genetic stability of the reprogrammed cells. The present work analyzes the hGMSC-derived iPS pluripotent genes’ expression in comparison to parental hGMSCs’ pluripotent genes’ expression. Since hGMSCs are multipotent cells that basically express SOX2, OCT4, and NANOG [23], it is interesting to study in what way the SOX2, OCT4, and NANOG expression changes when hGMSCs are reprogrammed. The real-time PCR and FACS results showed that the expression of all three pluripotent markers was much higher in hGMSC-derived iPS than in parental hGMSCs. SOX2, OCT4, and NANOG expression was confirmed by the immunofluorescence analysis. hGMSC-derived iPSs are more potent in stemness than hGMSCs. A very interesting detail to note is that the reprogramming method described in the current work not only determined the induction into pluripotency in the hGMSCs but also changed the expression of surface markers in hGMSC-derived iPSs, as shown by flow cytometric-analysis results. The typical mesenchymal markers CD29, CD44, CD73, CD90, CD105, CD146, and CD166, ref. [24] were decreased in hGMSC-derived iPS, confirming the loss of multipotency compared to the source cells. The pluripotency was demonstrated from the pluripotent markers previously described, and it was confirmed by CD117 expression, which resulted in being highly expressed in hGMSC-derived iPS and almost for nothing in hGMSCs. CD117 is expressed on both mouse and human embryonic cells [25], and it is fundamental for the adult organism development, so much so that a knockout of the relevant gene is incompatible with life [26]. CD117 data confirm that hGMSC-derived iPS have assumed a complete embryonic phenotype. Furthermore, CD117-positive stem cells selected from amniotic fluid have been shown to differentiate into cell lineages representing all three embryonic germ layers without generating tumors. However, the use of amniotic stem cells for tissue regeneration has limitations related to the heterogeneous phenotype and the reduced expansion potential of these cells [27]. Therefore, the CD117 expression, which was high in hGMSC-derived iPS and almost not present in the starting hGMSCs, makes hGMSC-derived iPS a promising alternative and better-performing candidate for regenerative medicine applications. A teratoma test in vivo was used to assess the pluripotency but mainly due to the lack of definitive clinical significance and economic expense; the in vitro embryonic body (EB) formation is an excellent alternative to the teratoma test for evidencing pluripotency in putative ES/iPS cells [28]. The EB is a three-dimensional aggregate of cells containing differentiating cell populations, which demonstrate that the ESs/iPSs under evaluation are capable of differentiating in the three embryonic layers and therefore potentially in all tissues [11]. As previously described, it is evident that many iPS cell lines that were in the process of being characterized, but at the end of the reprogramming, they expressed the pluripotency genes and proteins, and when these cells were used to constitute the EBs, they were not able to form them and to give origin to the three germinal layers. For this reason, they could not be considered actual embryonic cells. The iPS-derived hGMSCs not only proved to be pluripotent in morphology and pluripotency genes and proteins expression but, in vitro, are able to form EBs, as visible from the optical microscope. Furthermore, the real-time PCR analysis carried out on these EBs demonstrated that they were made up of three very distinct cell populations, which were highlighted by the positivity in PCR for the expression of markers TUBB3 (ectoderm), SOX17 (endoderm), and T (mesoderm), which were not expressed in the undifferentiated colonies starting of hGMSCs-derived iPS and in parental hGMSCs. The confocal analysis for the expression of TUJ1 (ectoderm), SOX17, and αSMA (mesoderm) markers confirmed the ability of the iPS-derived hGMSCs within the EBs to differentiate into ectoderm, mesoderm, and endoderm tissues, respectively. Furthermore, the three-dimensional EBs’ analysis allowed the specific localization of these germ populations within the EBs, while the cytochemical and immunofluorescence analysis on the EB sections allowed for the evaluation of the expression, morphology, and localization of cellular markers within individual cells. A further interesting result derives from the morphological evaluation of the population migrating from the EB transferred to a flat-well plate. Around the EB, the migrating cells form a compact colony wherein the distinction of the three germinal populations is maintained, and the cells migrate with a specific morphology and localization even when adhered to the plate, as if to indicate a cellular memory in the differentiation organization. The FACS results of EVs produced by both hGMSCs and hGMSC-derived iPS showed that hGMSC-derived iPS produced a larger number of vesicles than parental gingival cells. This could represent an advantage for the use of vesicles as a therapeutic strategy since, for potential EV therapies, the number of vesicles needed is high, and the new iPS line, by providing a greater quantity of vesicles, could be a cell line that is better-performing than parental hGMSCs to produce EV-based therapies. Furthermore, CD90, a typical gingival vesicles marker, greatly decreased in the vesicles of the related iPS, demonstrating that these pluripotent cells lost the multipotency phenotypic characteristics also at the level of the vesicular surface. The reprogrammed cells showed some changes in epigenetic pattern: EVs from hGMSC-derived iPS rearranged their miRNA contents compared to the parental cells. The miRNAs that were identified in both the hGMSC and hGMSC-derived iPS groups are differentially expressed: down- and up-expressed in hGMSC-derived iPS EVs compared to hGMSCs. Furthermore, not only did the parental cells’ vesicles contain 66 miRNAs that were not identified in iPS EVs, but, in iPS EVs, we identified 8 miRNAs that are not in parental ones. In addition to the data analyzed using the IPA functional analysis for understanding their involvement in different physio-pathological conditions, the IPA-generated network allowed us to observe that the central heart of the network identified for hGMSC EVs’ miRNAs is HNF4A, while for those identified in hGMSC-derived iPS EVs is TP53, so the EVs from the two groups could be used on different pathological targets in clinical studies. The eight miRNAs present in hGMSC-derived iPS EVs and not present in the vesicles of the parental line are hsa-miR-122, hsa-miR-138, hsa- miR-142-3p, hsa-miR-143, hsa-miR-152, hsa-let-7b, mmu-miR-451, and hsa-miR-486. Most of the eight miRNAs identified exclusively in hGMSC-derived iPS EVs are found to be tumor suppressors from various bibliographic studies [29]: miR-122 is involved in apoptosis through the inactivation of PI3K-Akt- mTOR signaling pathway, which may be lead to the up-regulation of TP53 expression [30], and it is related also with the suppression of Wnt/beta-catenin pathway [31]; miR-138 inhibits MYC expression and suppresses tumor growth [32]; miRNA-143 appears to be an indirect regulator of cancer glycolysis [33]; miR-152 could inhibit tumor cell proliferation and cell cycle progression by regulating the expression of KLF5 [34]; let-7b, when decreased, is often associated with cancer [35]; miR-451 directly targets Myc gene, repressing its expression and acting as a tumor suppressor [36]; and miR-486 regulates epithelial-to-mesenchymal transition and invasion of osteosarcoma cells through targeting PIM1 [37]. Some of these miRNAs also have an anti-inflammatory role. From bibliographic research, miR-138 appears to have an anti-inflammatory effect regulating the expression of NFkB p65 [38], let-7b targets 3’-UTR of tool-like receptor 4 (TLR4) mRNA [39], and miR-451 inhibits TLR4 expression [40]. Others members of these miRNAs are involved in the development of organs and tissues: has-miR-122 in the liver development [41]; miR-138 for osteogenesis process [42]; miR-142-3p in the formation of mesoderm-derived hemogenic endothelium [43]; miR-143 for cardiac development [44]; let-7b in neural differentiation [45] and mammals’ embryonic implantation [46]; miR-451 for erythropoiesis process [47]; and miR-486 for skeletal muscle development processes [48]. Therefore a decrease in miR-142-3p and miR-486 is revealed in cardiovascular diseases [49,50]. The EVs produced by hGMSC-derived iPS cells could be studied to evaluate anti-tumor, anti-inflammatory, and protective effects in different pathological conditions. It is important to underline that, despite the new research on iPS cells that has allowed us to increase the knowledge in the field of regenerative medicine, their use for clinical practices is still far away mainly because of the difficulty in translating the results into safe clinical applications. In addition to the use of iPS, future directions for potential clinical applications also include other methodologies, such as the transfer of mitochondria from stem cells to other cell types in order to provide fuel for their performance [51]. Furthermore, the constitution and analysis of directly patient-derived organoids and embryoid bodies represent an important tool for identifying new clinical insights, especially to predict drug response and discover new biomarkers [52].

## 4. Materials and Methods

### 4.1. hGMSCs’ Culture Establishment

Gingiva biopsy was picked up during oral surgery procedures in a patient who was to undergo orthodontic treatment. The recruited patient was in good health and had no oral diseases. After tissue collection, the gingival tissue fragments were crumbled and washed five times with phosphate-buffered saline (PBS) (Lonza, Basel, Switzerland). After these washes, the fragments were placed in a culture dish inside an incubator at 37 °C with a humidified atmosphere of 5% CO_2_ in the air. The culture medium used to incubate the fragments was MSCBM-CD (Lonza), a growth medium for chemically defined mesenchymal stem cells. The medium was changed every 2 days to stimulate the growth of hGMSCs. After 2 weeks of culture, hGMSCs spontaneously migrated from the explants into the plate. To assess cell morphology and the ability to adhere to a plastic substrate, hGMSCs at the second passage were seeded on Petri dishes and stained with a toluidine blue solution [53].

### 4.2. Flow Cytometry Analysis of hGMSC and hGMSC-Derived iPS Phenotypes

The analysis of cell phenotypes was carried out as previously reported [54]. Briefly, 3 × 10^5^ cells/sample were stained for the following surface antigens: CD29–phycoerythrin (PE), CD31–fluorescein isothiocyanate (FITC), CD34 PE–Cyanin7 (CY7), CD45 allophycocyanin H7 (APC-H7), CD73 PE, CD90 FITC, CD106 FITC, CD117 PE, CD144 FITC, CD146 PE, CD326 peridinin-chlorophyll-protein (PerCP)–Cy5.5, HLA-ABC FITC, HLA-DR PE (all from BD Biosciences, San Jose, CA, USA); CD44 FITC, CD105 FITC, CD166 FITC (Ancell Corporation, Stillwater, MN, USA), and CD133 APC (Miltenyi Biotec, Köln, Germany), for 30 min at 4 °C in the dark. The staining of intracellular antigens was carried out on at least 3 × 10^5^ cells/sample using the BD Cytofix/Cytoperm (cat. 554722. BD Biosciences, San Jose, CA, USA) to fix and permeabilize the membranes (20 min at room temperature (RT), in the dark). Samples were then incubated with the following list of antibodies: CD34-CY7, NANOG-PE, SOX2-FITC, OCT4-PE, SSEA-4-FITC (BD Biosciences, San Jose, CA, USA) for 30 min at 4 °C in the dark. Non-specific fluorescence was evaluated as established by current guidelines [55].

Finally, 104 cells were acquired by a FACS CantoII flow cytometer (BD Biosciences) and then analyzed by FACSDiva v8.0.3 (BD Biosciences) and FlowJo v10.9.0 (BD Biosciences) software. Instrument performances and data reproducibility were maintained and checked by using the Cytometer Setup and Tracking (CS&T) Module (BD Biosciences).

### 4.3. Colorimetric Detection of hGMSCs’ Mesenchymal Differentiation

Since hGMSCs are capable of differentiating into cells of mesenchymal lines, such as adipose and bone tissues, to confirm this property hGMSCs were differentiated in vitro and evaluated by colorimetric analysis and reverse-transcription polymerase chain reaction (RT-PCR). To achieve adipogenic differentiation, hGMSCs were grown in a 24-well plate at a density of 2 × 104 cells/well with 500 μL medium. For adipogenic differentiation culture, 10 mmol/L of dexamethasone, 10 nmol/L of 3-isobutyl-1-methylxanthine, 5 mg/mL of insulin, and 60 mmol/L of indomethacin were added to the MSCBM-CD medium for 28 days. The change in medium was performed every 3 days. To assess differentiation, Oil Red O solution (Sigma-Aldrich, Milan, Italy) was used to highlight lipid droplets at the cytoplasmic level. To start osteogenic differentiation, hGMSCs were grown in a 24-well plate at a density of 2 × 104 cells/well with 500 μL medium. Osteogenic differentiation was induced by adding 10 nmol/L of dexamethasone, 10 nmol/L of beta-glycerophosphate (Sigma-Aldrich), and 50 mmol/L of AA to MSCBM-CD medium, used for 21 days. Osteogenic differentiation was identified by staining with Alizarin Red S (Sigma-Aldrich), which highlights calcium deposits in osteogenic cells. After adipogenic and ostogenic staining, the differentiated hGMSCs were visualized under a Leica DM IL inverted light microscope (Leica Microsystems, Milan, Italy).

### 4.4. RT-PCR Analysis of hGMSCs’ Mesenchymal Differentiation

The differentiation between adipogenic and osteogenic lines, first detected by colorimetric assays, was confirmed by RT-PCR, evaluating the expression of specific osteogenic and adipogenic genes. Runt-related transcription factor-2 (RUNX-2) and alkaline phosphatase (ALP) were analyzed for osteogenic commitment, and fatty acid binding protein 4 (FABP4) and peroxisome proliferator-activated receptor γ (PPARγ) for adipogenic differentiation. RT-PCR was performed with the TaqMan Universal PCR Master Mix according to the protocol instructions (Applied Biosystems, Foster City, CA, USA). β-2 microglobulin (B2M Hs999907_m1) (Applied Biosystems) was used for normalization.

### 4.5. Reprogramming of hGMSCs

Reprogramming of the hGMSCs was based on the Stemgent StemRNA 3rd Gen Reprogramming Kit’s (Stemgent Cat. No. 00-0076) guidelines. From day 5, we saw only medium changes with Nutristem until the day when colony formation became evident (day 10 to day 14). If colonies have already formed by day 10, picking can begin.

### 4.6. hGMSC-Derived iPS Colonies Picking and Expansion

Colonies were formed on day 13. Before picking, an appropriate number of wells of a 12-well plate were coated with 0.5 mL of 2.4 μg/mL iMatrix-511 in PBS. After 1 h of incubation (37 °C), the iMatrix was replaced with mTeSR Plus (STEMCELL Technologies 100-0276) maintenance medium for hPSCs. At this point, the colonies from the mother plate were picked using the 20 μL pipette, and each colony was transferred to a different coated well to allow for the expansion. The colonies transferred from the mother plate in a new coated well were considered passage 1.

### 4.7. hGMSC-Derived iPS Optical Microscopy and Scanning Electron Microscope

The morphology of the cells and colonies was assessed by inverted light microscopy (Leica Microsystem), and the images were acquired by using light microscopy connected to a high-resolution digital camera DFC425B Leica (Leica Microsystem). For scanning electron microscope, hGMSC-derived iPS colonies were fixed for 1 h in 4% glutaraldehyde in 0.05 M phosphate buffer (pH 7.4). After the washes with cacodylate buffer, they were incubated for 30 min with osmium in a 1:1 ratio in cacodylate buffer. Then, we performed dehydration in increasing ethanol concentrations, and, afterward, a critical point dried. hGMSC-derived iPS colonies were then mounted on aluminum stubs and gold-coated in an Emitech K550 (Emitech Ltd., Ashford, UK) sputter-coater before imaging by means of a SEM (ZEISS, EVO 50, Jena, Germany).

### 4.8. Karyotype Reconstruction

Cytogenetic analysis was performed for the characterization of the hGMSC-derived iPS cell line from hGMSCs to demonstrate their chromosomal stability. hGMSC-derived iPS and hGMSCs’ adherent cultures were carried out in a sterile Petri dish with a glass slide (amniodish), with mTESR for hGMSC-derived iPS and MSCBM-CD for the hGMSCs. The cultures were treated for 1 h with colcemide 10 μg/mL in DPBS and subsequently processed with a 0.56% KCl solution and washed with Carnoy’s liquid (methanol/acetic acid 3/1). Metaphasic chromosomes were treated with GTG banding and reconstructed through Nikon Genikon software 4.0.

### 4.9. hGMSC-Derived iPS Colonies Immunofluorescence Analysis

The pluripotency of the colonies formed by the hGMSC-derived iPS was assessed by searching for the expression of pluripotency markers Oct4, Nanog, and Sox2 through immunofluorescence with confocal microscopy. The starting hGMSCs were used as control (CTRL). The hGMSC-derived iPS colonies were plated on cellview cell culture dish, glass bottom, treated before with Geltrex (Gibco A1569601). The starting hGMSCs were plated on cellview cell culture dish, glass bottom. Once at confluence, hGMSC-derived iPS colonies and CTRL were fixed for 1 h with 4% paraformaldehyde in 0.1 M of PBS (pH 7.4) (Lonza, Basel, Switzerland) at room temperature. Successively, several washes were performed. After the washes, the permeabilization of hGMSC-derived iPS colonies was conducted with 0.5% Triton X-100 in PBS buffer (Lonza) for 6 min and stopped with rapid washes in PBS. Then, the saturation was performed with 5% skimmed milk in PBS for 2 h. Incubation with primary antibodies was overnight with anti-human Oct4 (1-2 μg/mL, mouse) (Invitrogen), anti-human Sox2 (1:100, rabbit) (Cell Signaling), and anti-human Nanog (1:200, rabbit) (Cell Signaling). The following day, incubation with secondary antibodies was performed for 1 h at 37 °C, using Alexa Fluor 568-conjugated goat anti-rabbit secondary antibodies (1:200; ThermoFisher, Life Technologies, Monza, Italy) to detect Nanog and Sox2 and Alexa Fluor 488-conjugated goat anti-mouse secondary antibodies (1:200; Thermo Fisher, Life Technologies, Monza, Italy) to visualize Oct4. To dye the nuclei, the samples were dyed with TOPRO (1:200) (T3605, Molecular Probes) for 1 h. Since Nanog alone was not sufficient as an indicator of pluripotency, so its co-expression with Oct4 in the same sample was analyzed. Actin was highlighted with Alexa Fluor 488 phalloidin green fluorescent conjugate (1:200, incubation for 1 h) (A12379, Molecular Probes) only in the sample in which Sox2 expression was evaluated, for emission spectra reasons. The Zeiss LSM800 confocal system (Carl Zeiss, Jena, Germany) was utilized to obtain images.

### 4.10. DNA Finger Printing

Genomic DNA from both hGMSCs and hGMSC-derived iPS was extracted using Nucleospin miRNA and RNA/DNA buffer set kits (Macherey-Nagel, Milan, Italy) according to the manufacturer’s instructions. Quantity and quality of DNA were assessed by Qubit 2.0 (Invitrogen, Monza, Italy). The amplification of the 6-dye multiplex PCR-CE-based AmpFLSTR^®^ Yfiler^®^ Plus PCR Amplification Kit (Thermo Fisher Scientific, Waltham, MA, USA) was performed in a single multiplex PCR reaction (25 μL in total, containing 10 μL master mix, 5 μL primer mix, and 10 μL genomic DNA) on SimpliAmp 96-Well Thermal Cycler System (Thermo Fisher Scientific, Waltham, MA, USA), following the manufacturer’s instructions. Amplified products were separated by capillary electrophoresis (CE) on a 3500xL Genetic Analyzer (Thermo Fisher Scientific, Waltham, MA, USA).

### 4.11. Formation of Embryoid Bodies (EBs)

Cells cultures in a 6-well plate at a confluence of 85% were used to plate in 96U bottom plate (Thermo scientific 312303), forming 96 embryoid bodies (approximately 5000 cells per well). The medium for the embryoid bodies (EBs) was established as follows: Kncokout DMEM 485 mL (Gibco 10829-018), MEM non-essential amino acids 5 mL (Gibco 11140-050), L-glutamine 5 mL (Sigma G7513), Pen/Strep 5 mL (EuroClone ECB3001D), and β-mercaptoethanol 350 μL (Sigma M-7154). The EBs’ culture medium was changed once a day until the sixteenth day of culture. On day 8, the EBs were transferred in cell culture dishes, glass bottom (CELLviewTM), and cultured until the sixteenth day to evaluate the migrant population. On day 16, the EBs were used for PCR analysis and for immunofluorescence analysis.

### 4.12. qPCR

RNA was extracted from the three samples, hGMSCs, hGMSC-derived ips, and EBs, using PureLink RNA Mini kit (Invitrogene 12183018A). RNA was quantified through UV-VIS spectrophotometer with BioSpectrometer (Eppendorf, Hamburg, Germany). Subsequently, the cDNA of the three samples was obtained by applying the protocol of High-Capacity cDNA Reverse Transcription kit (Applied biosystems 4368814). The cDNA obtained was processed in real-time PCR with the following primers: NANOG (IDT Reference #: 234091833), SOX2 (IDT Reference #: 234104100) and POU5F1 (OCT4) (IDT Reference #: 234104096) as pluripotency genes, TUBB3 (IDT Reference #: 234091845), SOX17 (IDT Reference #: 234091841), and T (IDT Reference #: 234091837), for the three germinal differentiation markers ectoderm, endoderm, and mesoderm, respectively. GAPDH (IDT Reference #: 234104092) was used as a housekeeping gene.

### 4.13. EB Sections’ Histochemistry and Immunofluorescence Staining

EBs were processed for histochemistry staining to study the morphological features. EBs were washed with PBS and fixed with 4% paraformaldehyde in 0.1 M of PBS (pH 7.4) for 1 h and 30 minutes. Successively, they were included in LR white resin, cut on the microtome, and stained with toluidine blue. Two immunofluorescence detection sections samples were permeabilized with 0.5% Triton X-100 in PBS buffer (Lonza) for 10 min and stopped with rapid washes in PBS. Samples were saturated with 5% skimmed milk in PBS for 2 h. Sections were incubated overnight with primary antibodies with anti-human Sox17 (1:500, goat) (R&D system AF1924), anti-human αSMA (1:500, mouse) (Sigma A2547), and anti-human Tuj1 (1:300, mouse) (Stem cell technologies #60052). Then, the following secondary antibodies were used for 1 h at 37 °C: Alexa Fluor 488-conjugated monkey anti-goat secondary antibodies (1:200; ThermoFisher, Life Technologies, Monza, Italy) for Sox17; and Alexa Fluor 488-conjugated goat anti-mouse secondary antibodies (1:200; Thermo Fisher, Life Technologies, Monza, Italy) for αSMA and Tuj1. TOPRO (1:200 for 1h incubation) (T3605, Molecular Probes) was used for the nuclei. Alexa Fluor 568 phalloidin red fluorescent conjugate (A12379, Molecular Probes) (1:200, incubation for 1 h) highlighted cytoskeleton actin. EB sections were analyzed by means of the confocal laser scanning microscopy, LSM800 confocal system (Carl Zeiss, Jena, Germany).

### 4.14. Immunofluorescence Analysis of 3D EBs and EBs Migrating Population

The EBs’ morphology and the differentiation in the three germ lines were assessed by searching for the expression of ectodermal marker beta-tubulin III (Tuj1), mesodermal marker Alpha-Smooth Muscle Actin (αSMA), and endodermal marker SRY-Box Transcription Factor 17 (Sox 17) through immunofluorescence with confocal microscopy. The 3D EBs and the EBs transferred on cell culture dishes, glass bottom (CELLviewTM), with migrating population, were fixed for 1 h and 30 minutes with 4% paraformaldehyde in 0.1 M of PBS (pH 7.4) (Lonza, Basel, Switzerland) at room temperature. Successively, several washes were performed. After the washes, the permeabilization was conducted with 0.5% Triton X-100 in PBS buffer (Lonza) for 10 min and stopped with rapid washes in PBS. Then, the saturation was performed with 5% skimmed milk in PBS for 2 h. Incubation with primary antibodies was performed overnight with anti-human Sox17 (1:500, goat) (R&D system AF1924), anti-human αSMA (1: 500, mouse) (Sigma A2547), and anti-human Tuj1 (1:300, mouse) (Stem cell technologies #60052). The following day, incubation with secondary antibodies was performed for 1 h at 37 °C, using Alexa Fluor 488-conjugated monkey anti-goat secondary antibodies (1:200; ThermoFisher, Life Technologies, Monza, Italy) to detect Sox17 instead of Alexa Fluor 488-conjugated goat anti-mouse secondary antibodies (1:200; ThermoFisher, Life Technologies, Monza, Italy) to visualize αSMA and Tuj1. To dye the nuclei, the samples were dyed with TOPRO (1:200) (T3605, Molecular Probes) for 1 h. Actin was highlighted with Alexa Fluor 568 phalloidin red fluorescent conjugate (1:200, incubation for 1 h) (A12379, Molecular Probes). The Zeiss LSM800 confocal system (Carl Zeiss, Jena, Germany) was utilized to obtain images.

### 4.15. EVs Isolation and Quantification

EVs were isolated from the culture medium of hGMSCs and hGMSC-derived iPS cultured for 48 h. Then, 10 mL of the supernatants was collected and processed using the ExoQuick-TC (System Biosciences, Euroclone SpA, Milan, Italy), following the manufacturer’s protocol. A total of 2 mL of ExoQuick TC solution was added to the 10 mL of the supernatant. The mix was incubated overnight at 4 °C, without rotation. The samples were centrifugated at 1500× *g* for 30 min to sediment the EVs, and the pellets were resuspended in 500 μL PBS. An aliquot was used to perform the quantization with BCA protein assay (Thermo Scientific, BCA Protein Assay Kits Catalog number: 23225).

### 4.16. EVs Characterization by FACS

EVs were identified, analyzed, and enumerated in cell supernatants. For each sample, 1 µL of lipophilic cationic dye (LCD; Becton Dickinson-BD Biosciences, San Jose, CA, USA; Catalog, #626267, Custom Kit), a pan-EV dye, and 0.2 µL of Phalloidin FITC-conjugated (BD Biosciences, Catalog, #626267, Custom Kit) was added to 100 μL of cell supernatants. After 45 min of staining (RT, in the dark), 400 μL of PBS 1X was added to each tube, and 1 × 106 events/sample were acquired by flow cytometry (FACSVerse, BD Biosciences, San Jose, CA, USA; three lasers, eight-color configuration), placing the trigger threshold on the lipophilic cationic dye channel (APC) [21]. Amplifier settings for forward scatter (FSC) and side scatter (SSC), as well as for any fluorescence channel, were set in logarithmic mode, and all parameters were visualized as height (H) signal.

Instrument performances, data reproducibility, and fluorescence calibrations were monitored by the Cytometer Setup and Tracking Module (BD Biosciences). The evaluation of non-specific fluorescence was obtained by acquiring FMO controls. Data were analyzed using FACSuite v1.0.6.5230 (BD Biosciences) and FlowJo v10.9.0 (BD Biosciences) software. EVs’ concentrations were calculated based on the volumetric count function.

### 4.17. EVs miRNAs Extraction

Total RNA extraction, including microRNAs (miRNAs), from EVs obtained by hGMSCs and hGMSC-derived iPS was manually isolated using the Total Exosome RNA and Protein Isolation Kit (Invitrogen, Thermo Fisher Scientific, Waltham, MA, USA) according to the manufacturer’s instructions. Quantity and quality of total RNA were assessed by microvolume UV–Vis spectrophotometer NanoPhotometer (Implen, GmbH, Munich, Germany).

### 4.18. miRNAs’ Characterization Profile

MicroRNA analysis in EVs retrieved from hGMSCs and hGMSC-derived iPS was carried out by TaqMan™ Array Human MicroRNA A Cards v2.0 (Applied Biosystems, Foster City, CA, USA). An initial Megaplex RT Re211 action was performed in order to reverse-transcribe miRNAs, using the Megaplex RT primer Pool A and TaqMan MicroRNA Reverse Transcription kit (Applied Biosystems, Waltham, MA, USA). Afterwards, a preamplification reaction of complementary DNAs (cDNAs) was also executed through use of the TaqMan PreAmp Master Mix 2X and Megaplex PreAmp Primers pool A (Applied Biosystems, Waltham, MA, USA). The pre-amplified products were then diluted with 0.1X TE at pH 8.0 and subsequently mixed with the TaqMan™ Universal Master Mix II, no UNG (Applied Biosystems, Waltham, MA, USA) and nuclease-free water. Finally, they were loaded into the TaqMan™ Array Human MicroRNA A plates and processed by qRT- PCR, using a QuantStudioTM 7 Pro Real-Time PCR detection system (Life Technologies, Carlsbad, CA, USA).

### 4.19. IPA Functional and Biological Network Analysis

MiRNAs were analyzed by Ingenuity Pathway Analysis 24.0.1 software in order to highlight the principal functions, cellular processes, and molecular networks in which they actively participate. An IPA-inferred network analysis was also generated for the miRNAs identified from the EVs obtained from the parental line (hGMSCs) and the reprogrammed line (hGMSC-derived iPS), highlighting the miRNAs in common and not between the two cell lines, as well as for those differentially expressed between the two cell lines. The mechanistic network of these molecules was evidenced based on their connectivity and enrichment statistics.

### 4.20. Statistical Analysis

Statistical significance was established with GraphPad 5 (GraphPad, San Diego, CA, USA) software utilizing one-way ANOVA, followed by post hoc Tukey’s multiple comparisons analysis. Values of *p* < 0.05 were considered statistically significant.

## 5. Conclusions

A new iPS cell line was established that reprograms, for the first time, primary human gingival mesenchymal stem cells through a non-integrating method. The iPS-derived hGMSCs obtained proved to be pluripotent in all aspects, as in the expression of pluripotency markers, in the constitution of the embryoid bodies’ structure, and in the three germ-layer differentiation, thus overcoming the limit of many other iPS cell lines which were incapable of reaching the embryoid body stage despite expressing pluripotency markers. Furthermore, in the present work, it was evidenced that, compared to the parental hGMSCs, the new iPSs also rearranged the membrane receptors and the miRNA content of their EVs. Despite the limitations associated with the use of iPS in potential clinical applications, the hGMSC-derived iPS cell line not only represents a new alternative source of pluripotent cells for the study of adult tissues regeneration, but it is also a resource of EVs which, having different potential compared to those produced by other cells, provide potential new cell-free therapeutic approaches bypassing the instability identified in the use of iPS lines for clinical purposes. 

## Figures and Tables

**Figure 1 ijms-25-09169-f001:**
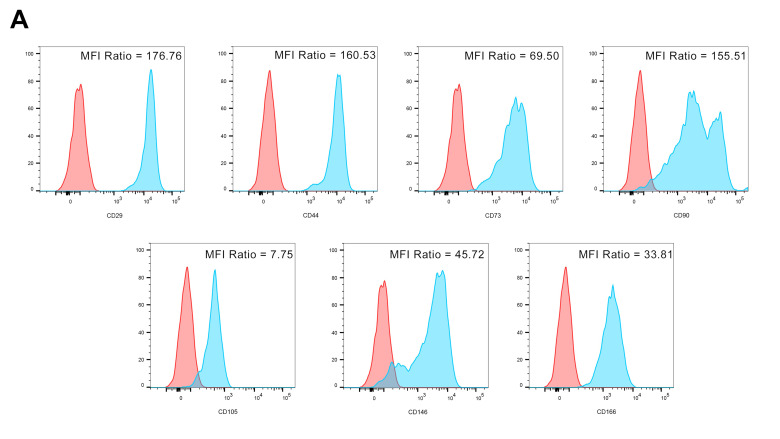
Cytofluorimetric analyses. (**A**) A list of markers expressed by human gingival mesenchymal stem cells (hGMSC)-positive markers analyzed by flow cytometry. The overlayed histograms represent the expression of CD29, CD44, CD73, CD90, CD105, and CD146 (blue histograms) with respect to their controls (red histograms). Mean Fluorescence Intensity (MFI) ratios (the ratio between the MFI of the marker expression and the related autofluorescence) were also reported for each marker in the related histogram. (**B**) CD45, HLA-DR, and CD31 surface expressions were analyzed by flow cytometry both on hGMSCs and hGMSC-derived induced pluripotent stem cells (iPS) samples. The overlayed histograms represent the surface expression of CD45, HLA-DR, and CD31 (blue histograms) with respect to their controls (red histograms). (**C**) Phenotype of hGMSCs and their derived iPS. Typical markers used to demonstrate the pluripotency phenotype of iPSCs were analyzed by flow cytometry. The overlayed histograms represent the expression of NANOG, OCT3/4, SOX2, and SSEA4 (blue histograms) with respect to their controls (red histograms). Mean Fluorescence Intensity (MFI) ratios (the ratio between the MFI of the marker expression and the related autofluorescence) were also reported for each marker in the related histogram. (**D**) Expression of CD29, CD44, CD73, CD90, CD105, CD146, CD166 and CD117 (blue histograms) with respect to their controls (red histograms) in hGMSCs and hGMSC-derived iPS samples. Mean Fluorescence Intensity (MFI) ratios (the ratio between the MFI of the marker expression and the related autofluorescence) were also reported for each marker in the related histogram.

**Figure 2 ijms-25-09169-f002:**
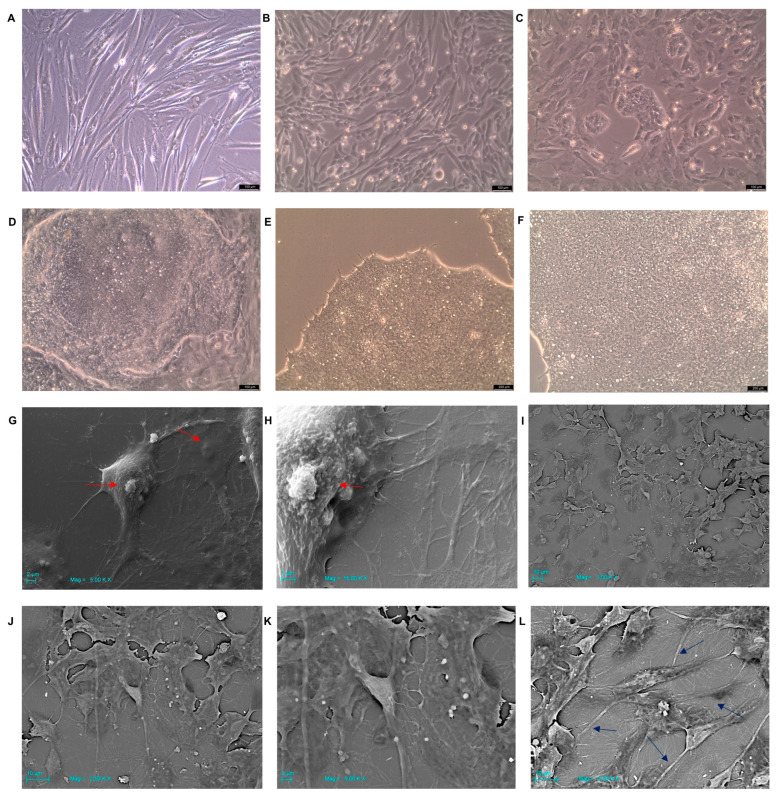
hGMSCs and iPS morphological features. (**A**) hGMSCs culture in basal conditions and (**B**) hGMSCs during reprogramming process at day 2 observed at inverted light microscopy. Mag: 10X. (**C**), Formation of the first iPS clusters on the eighth day of reprogramming. Optical microscopy. Mag: 10X. (**D**) Colonies’ formation at day 14 of reprogramming. Optical microscopy. Mag: 10X. (**E**,**F**) iPS cells’ morphology inside a colony after picking. Optical microscopy. Mag: 20X. (**G**,**H**) Scanning electron microscopy. hGMSC-derived iPS cell morphology with high nucleus/cytoplasm ratio. Red arrows: nucleus. (**G**) Mag = 5.00 KX. (**H**) Mag = 15.00 KX. (**I**–**K**) Scanning electron microscopy. hGMSC-derived iPS colonies’ morphology. (**I**) Mag = 1.00 KX. (**J**) Mag = 2.50 KX. (**K**) Mag = 5.00 KX. (**L**) Scanning electron microscopy. Blue arrows: cytoplasmic contacts of hGMSC-derived iPS cell morphology. Mag = 2.50 KX.

**Figure 3 ijms-25-09169-f003:**
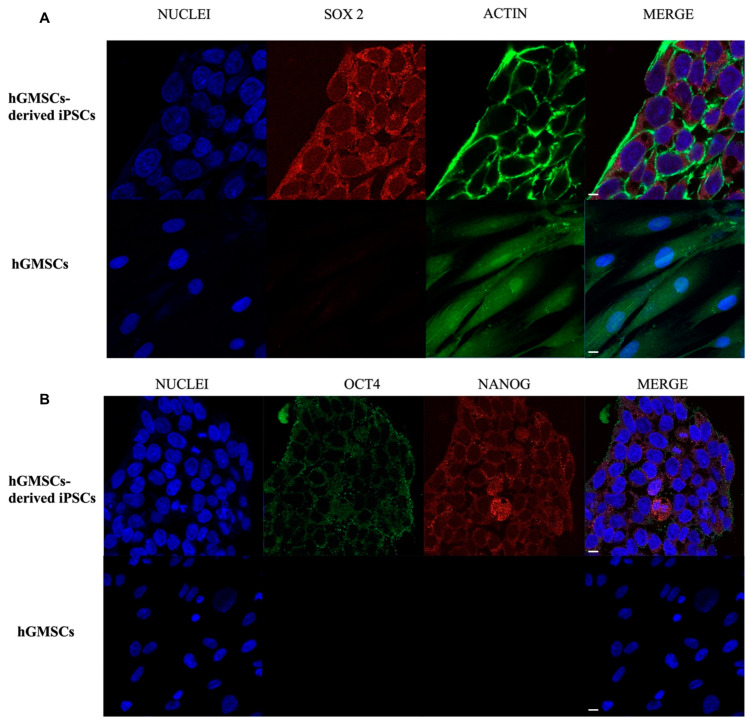
iPS colonies immunofluorescence detection. (**A**) SOX2 expression in hGMSC-derived iPS and in hGMSCs. Immunofluorescence. Confocal microscopy. Mag: 40X. (**B**) OCT4 and NANOG colocalization expression in hGMSC-derived iPS and in hGMSCs. Immunofluorescence. Confocal microscopy. Magnification (Mag): 40X.

**Figure 4 ijms-25-09169-f004:**
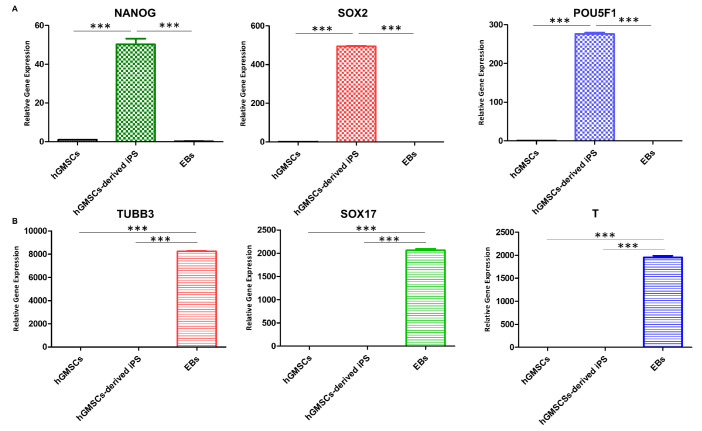
mRNA expression. (**A**) Real-time PCR data for the pluripotency genes’ expression in hGMSCs, hGMSC-derived iPS, and EB samples. (**B**) Real-time PCR data for the three germ layers’ genes’ expression in hGMSCs, hGMSC-derived iPS, and EBs samples. *** *p* < 0.001.

**Figure 5 ijms-25-09169-f005:**
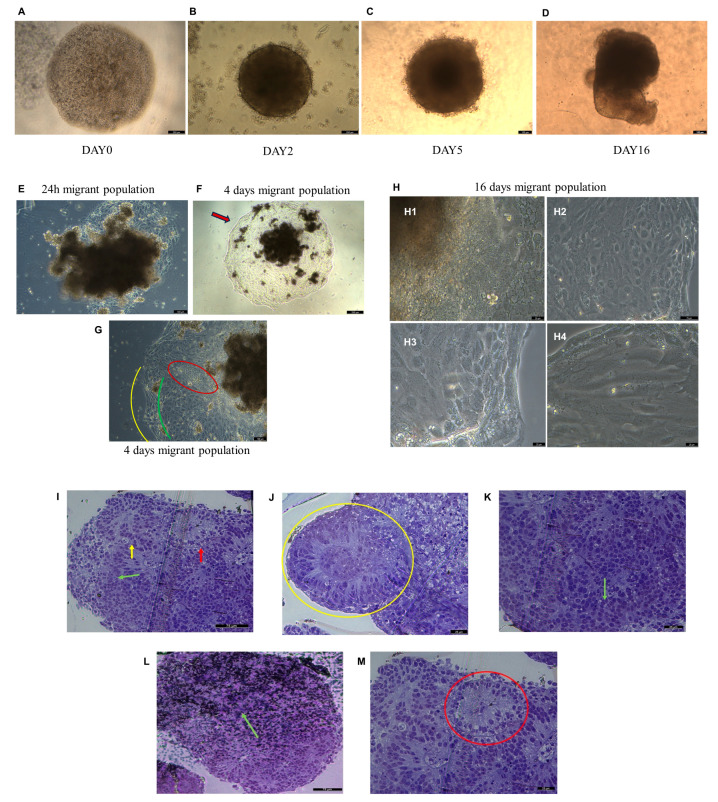
Embryoid bodies’ (EBs) morphological analyses. (**A**) iPS cells’ aggregation at day 0 for EBs formation. Optical microscopy. Scale bar: 250 µm. (**B**) EB on day 2. Optical microscopy. Scale bar: 250 µm. (**C**) EB on day 5. Optical microscopy. Scale bar: 100 µm. (**D**) EBs at day 16. Optical microscopy. Scale bar: 100 µm. (**E**) Migrant population from EBs after 24 h of EB transfer. Optical microscopy. Scale bar: 100 µm. (**F**) Migrant population from EBs after 4 days of EB transfer. Red arrow: distinct outline formation. Optical microscopy. Scale bar: 250 µm. (**G**) Migrant population from EBs after 4 days of EB transfer. The three areas differing in morphology and cellular organization are evidenced in red, yellow, and green. Optical microscopy. Scale bar: 100 µm. (**H**) Morphology of the three areas of different morphology after 16 days of EB transfer, already indicated after 4 days of EB transfer in (**G**) at lower magnification. Optical microscopy. (**H1**) Scale bar: 25 µm. (**H2**) Scale bar: 75 µm. (**H3**,**H4**) Scale bar: 25 µm. (**I**) Optical microscopy. EB’s section. Toluidine blue. Scale bar: 75 µm (yellow arrow, ectodermal tissue; green arrow, mesodermal tissue; and red arrow, endodermal tissue). (**J**) Optical microscopy. EB’s section. Toluidine blue. Scale bar: 25 µm (Yellow circle: neuroepithelial structure example). (**K**) Optical microscopy. EB’s section. Toluidine blue. Scale bar: 25 µm (green arrow: mesodermal tissue cells). (**L**) Optical microscopy. EB’s section. Toluidine blue. Scale bar: 75 µm (green arrow: connective fibers). (**M**) Optical microscopy. EB’s section. Toluidine blue. Scale bar: 25 µm (red circle: endodermic cells).

**Figure 6 ijms-25-09169-f006:**
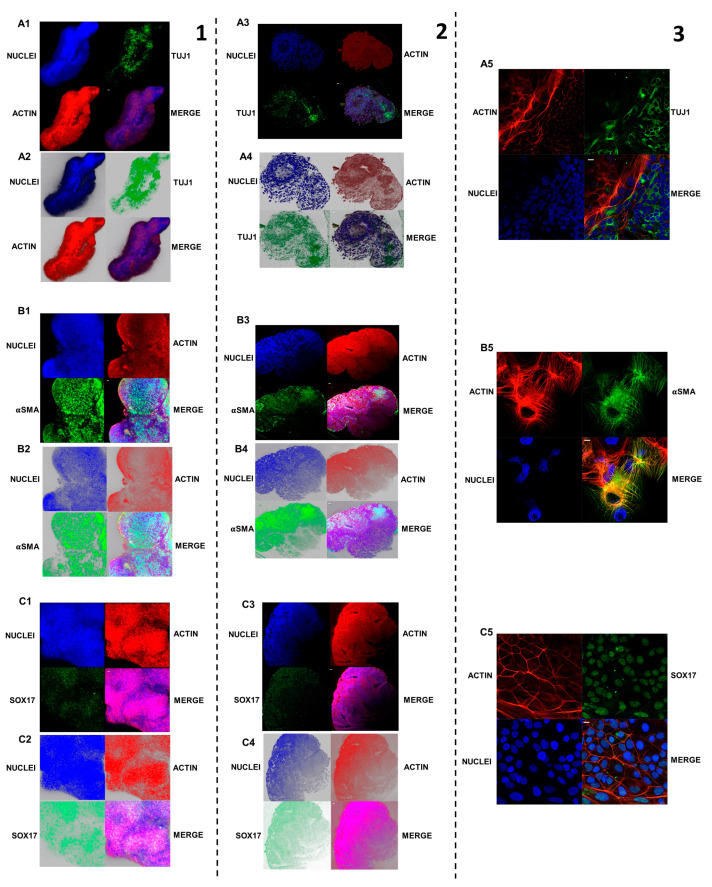
Germ layer differentiation markers in embryoid bodies. 1, Germ layer differentiation markers in 3D embryoid bodies. 2, Germ layer differentiation markers in embryoid bodies sections. 3, Germ layer differentiation markers in migrating population from embryoid bodies. (**A1**–**A5**), Expression of Tuj1 in EBs. Immunofluorescence. Red: Actin; Blu: Nuclei; Green: Tuj1. Confocal microscopy. (**A1**), 3D EBZ-stack Black background analysis. Mag: 20X. (**A2**), 3D EB Z-stack White background analysis. Mag: 20X. (**A3**), EBs sections immunofluorescence for tuj1. (blue: nuclei; red: actin; green: Tuj1). Confocal microscopy. Image acquisition with black background. Mag: 40X. (**A4**), EBs sections immunofluorescence for tuj1 (blue: nuclei; red: actin; green: Tuj1). Confocal microscopy. Image acquisition with black background. Image acquisition with white background. Mag: 40X. (**A5**), Tuj1-positive cells’ morphology of EB’s migrating population. Immunofluorescence. Red: actin; blue: nuclei; green: Tuj1. Confocal microscopy, Mag: 40X. (**B1**–**B5**), Expression of αSMA in EBs. Immunofluorescence. Red: actin; Blu: nuclei; green: αSMA. Confocal microscopy. (**B1**), 3D EB Z-stack. Black background analysis. Mag: 20X. (**B2**), 3D EB z-stack. White background analysis. Mag: 20X. (**B3**), EBs sections immunofluorescence for αSMA (blue: nuclei; red: actin; green: αSMA). Confocal microscopy. Image acquisition with black background. Mag: 40X. (**B4**), EBs sections immunofluorescence for αSMA (blue: nuclei; red: actin; green: αSMA). Confocal microscopy. Image acquisition with white background. Mag: 40X. (**B5**), αSMA positive cells morphology in migrating population from EBs. Immunofluorescence. Red: actin; blue: nuclei; green: αSMA. Confocal microscopy. Mag: 40X. (**C1**–**C5**), Expression of SOX17 in EBs. Immunofluorescence. Red: actin; blue: nuclei; green: SOX17. Confocal microscopy. **C1**), 3D EB Z-stack. Black background analysis. Mag: 20X. (**C2**), 3D EB Z-stack. White background analysis. Mag: 20X. (**C3**), EBs sections immunofluorescence for SOX17 (blue: nuclei; red: actin; green: αSMA). Confocal microscopy. Image acquisition with black background. Mag: 40X. (**C4**), EBs sections immunofluorescence for SOX17 (blue: nuclei; red: actin; green: αSMA). Confocal microscopy. Image acquisition with white background. Mag: 40X. (**C5**), Sox17 expression and positive cells morphology. Immunofluorescence. Red: actin; blu: nuclei; green: SOX17. Confocal microscopy Mag: 40X.

**Figure 7 ijms-25-09169-f007:**
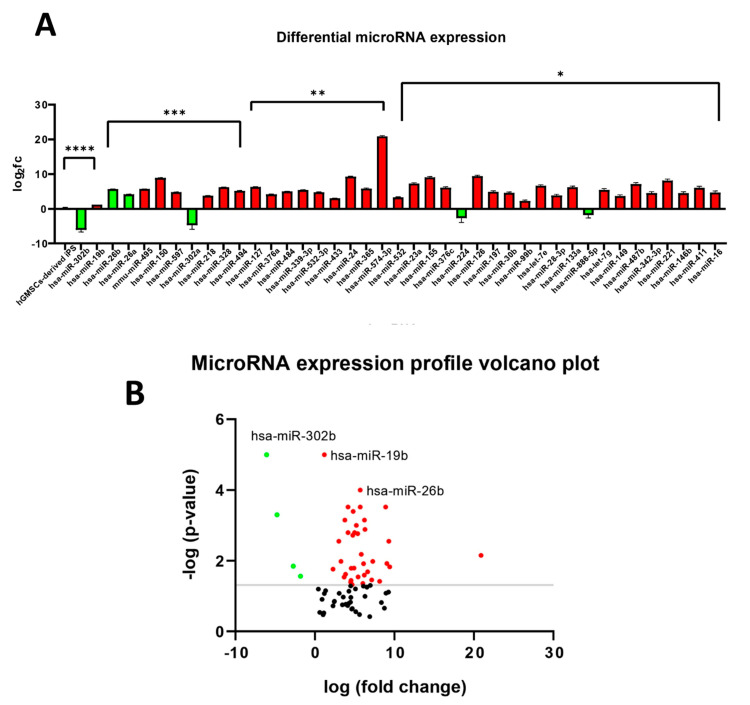
MicroRNA expression analysis in the parental hGMSCs extracellular vesicels (EVsEBs and hGMSC-derived iPS EVs. (**A**) Differential significant microRNA expression in the EVs of the reprogrammed compared to the parental line groups. (**B**) Volcano plot of the microRNAs modulated in the reprogrammed EV group compared to the parental EV group. In red are those up-regulated; in green are those down-regulated; while in black, below the −log (*p*-value) threshold, are the non-significantly modulated. The 3 named reported microRNAs resulted in being the most significantly modulated in this specific cohort. (**C**) IPA functional analysis of microRNAs characterized exclusively in the EVs of the parental line. (**D**) IPA functional analysis of microRNAs characterized exclusively in the EVs of the reprogrammed line. (**E**) IPA functional analysis on the cohort of significantly modulated microRNAs between the two studied groups. (**F**) Top IPA-generated network for microRNAs characterized in the parental EVs. (**G**) Second available IPA network underlining direct relationships between the characterized microRNAs in the parental-derived EVs. *p*-value **** < 0.0001, *** < 0.001, ** < 0.01 and * < 0.05.

## Data Availability

All data generated or analysed during this study are included in this published article and its Appendix A.

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
