# Peer review of "Autologous hGMSC-Derived iPS: A New Proposal for Tissue Regeneration"

_ijms, 2024, doi:10.3390/ijms25179169_

Round 1

Reviewer 1 Report

Comments and Suggestions for Authors

Dear authors

Below are my comments.

P2 lane 82 and P15 lane 523 : put the references in the form of a number as in the whole text.

P9 lanes 282-283. P17 lanes 582, 596 and 606-607. P18 lanes 662-663. P20 lane 744 :  change intervals

Materials and Methods are well described

P17 lane 580 Karyotype instead of Karyotipe in the title

Results

P3 lane 113 they were maintained

Figure 1 : the figure should be redone, the annotations on the diagrams are illegible. Indicate hGMSC instead of MSC.

P5 lane 169 : you introduce a new abbreviation hPSCs ?

Figure 2 : rather than putting Mag indicated the size by a scale bar on each photo. The photos should be brightened for better visualization. Indicate on the photos the days when the cells are collected. It would be better to add in an insert a higher magnification to show the morphology of the cells for E and F. For G and H indicate the nucleus. Indicate that K is an enlargement of J and H an enlargement of G. In L you write "cytoplasmic contacts..." can you show it because we can't see anything.

Figure 3 : the scale bar is missing. OCT4 in B is weak, we can't see it on the merge anymore. The intensity of the blue nuclei should be slightly reduced. Put the protein names in capital letters.

Figure 4 : indicate the p-value. Why is there 3* in A between hGMSC and EB when the expression is practically zero in both cases ? How many times have you done the experiment in triplicate? there is no error bar on the histograms.

P7 lane 238 : add red arrow after Fig. 5F

P8 lane 247 : add it in the sentence

Figure 5 : indicate the time on each photo for better reading. Lighten photo D a little. Redo the scale bars that are not visible. Indicate on H1 with arrows for example where the areas H2, H3 and H4 are. Did you use the same color code in G and I ?

Lane 259 : say what the 3 zones correspond to. We don't see much on G.

P9 lane 287 : you write "the actin showed..." but it would be necessary to prove that these are neural cells, morphology is not enough.

Figure 6 : The photos are very small and not of great quality. If you enlarge you can see the pixels. The quality of the photos should be improved. Put scale bars because the indicated magnifications seem wrong to me. Indicate the size of the EBs. Review the entire legend of the figure because it is not clear. In A5 in which area are we located in the sample ? In B the indicated magnifications are not possible because B5 is larger than the other Bs, which is why a scale bar is more indicated. In C5 SOX 17 is weak and we do not see it on the merge because the blue is strong. So rather do the merge ACTIN/SOX17. Write the caption for Figure 6 using the same character size.

Figure 7 : Figure to be reviewed. It is illegible, the characters are too small. In A and B the characters are not the same size. When you enlarge C and D it is blurry. You can't read anything in F and G.

P13 lanes 394 to 398 : This is not obvious with the results presented.

P14 lanes 427 to 430 : the sentence is really too long

Figure S1 : indicate calcium deposits in B with arrows. Put a visible scale bar because it is too small in C.

Figure S2 : put A and B in place of a and b

Figure S3 to put in bold. A, B and C instead of a, b, c

Figure S4 : Add A and B

Best regards

Author Response

Thank you to the Reviewer for his comments. Please see the attachment.

Reviewer 2 Report

Comments and Suggestions for Authors

Specifically:

The abstract provides a good summary of the research, including the problem being addressed, the methods used, and the main findings. However, it could benefit from more clarity, particularly in the description of the comparative study between parental hGMSCs and derived iPS cells.

The introduction could be more focused, especially when discussing the broader aspects of regenerative medicine. Some paragraphs could be condensed to maintain reader engagement. Additionally, the introduction would benefit from a clearer statement of the research hypothesis.

The methods section could benefit from a clearer structure, particularly by separating the different experimental procedures more distinctly. For example, the flow cytometry analysis and the characterization of extracellular vesicles (EVs) could be more clearly delineated. Additionally, a more explicit description of the statistical analyses performed would strengthen the section.

While the results are comprehensive, they are somewhat difficult to read due to the density of the information presented. Breaking down the results into more sections or subsections could improve readability. The discussion of EVs, in particular, could be expanded to better integrate it with the overall narrative of the paper.

The discussion could benefit from a more critical evaluation of the limitations of the study, such as the potential challenges in translating the findings to clinical applications. Additionally, some speculative statements regarding the potential uses of the iPS line could be more cautiously framed. In addition, some aspects, for future directions, could also be taken from recent papers regarding the transfer of mitochondria from stem cells into other cells types as a fuel for their performancies (see J Exp Clin Cancer Res. 2024 Jun 14;43(1):166. doi: 10.1186/s13046-024-03087-8. PMID: 38877575), as well as the use of other insights and analyses for organoids or embryonic bodies (see J Exp Cain Cancer Res 2023 Jan 6; 42 :8 dot 10.1186/s13046-022-02591-z PMID: 33604765).

The conclusion could be more concise and focused on the key takeaways from the study. It should also briefly mention any future directions or unresolved questions that arose from the research

Comments on the Quality of English Language

Minor editing of English required.

Author Response

Thanks to the reviewer for the detailed comment and for his help in improving the considered paper. The reviewer's suggestions have been followed to revise the manuscript. The updated paper file has been uploaded according with the reviewer.